# Impact of the COVID-19 pandemic and multiplex polymerase chain reaction test on outpatient antibiotic prescriptions for pediatric respiratory infection

Daisuke Kitagawa[1,2]*, Taito Kitano[3]*, Madoka Furumori[1], Soma Suzuki[1], Yui Shintani[1], Hiroki Nishikawa[4], Rika Suzuki[4], Naohiro Yamamoto[4], Masayuki Onaka[4], Atsuko Nishiyama[4], Takehito Kasamatsu[5], Naoyuki Shiraishi[5], Yuki Suzuki[2], Akiyo Nakano[2], Ryuichi Nakano[2], Hisakazu Yano[2], Koichi Maeda[5], Sayaka Yoshida[4], Fumihiko Nakamura[1]

1 Department of Laboratory Medicine, Nara Prefecture General Medical Center, Nara, Japan, 2 Department of Microbiology and Infectious Diseases, Nara Medical University, Kashihara, Nara, Japan, 3 Johns Hopkins Bloomberg School of Public Health, Baltimore, Maryland, United States of America, 4 Department of Pediatrics, Nara Prefecture General Medical Center, Nara, Japan, 5 Department of Infectious Diseases, Nara Prefecture General Medical Center, Nara, Japan

* d.kitagawa.med@gmail.com (DK); taito.kitano.0110@gmail.com (TK)

**Data Availability Statement:** "The data that support the findings of this study are included in

## Abstract

This study aimed to evaluate the impact of the prolonged COVID-19 pandemic on outpatient antibiotic prescriptions for pediatric respiratory infections at an acute care hospital in Japan in order to direct future pediatric outpatient antibiotic stewardship. The impact of the COVID-19 pandemic and the FilmArray Respiratory Panel (RP) on outpatient antibiotic prescriptions was assessed from January 2019 to December 2021 using an interrupted time series analysis of children <20 years. The overall antimicrobial prescription rate decreased from 38.7% to 22.4% from the pre-pandemic period to the pandemic. The pandemic (relative risk [RR] level, 0.97 [0.58–1.61]; P = 0.90; RR slope, 1.05 [0.95–1.17] per month; P = 0.310) and FilmArray RP (RR level, 0.90 [0.46–1.75]; P = 0.75; RR slope, 0.95 [0.85–1.06] per month; P = 0.330) had no significant effect on the monthly antibiotic prescription rates. The COVID-19 pandemic was not significantly related to the antibiotic prescription rate, suggesting that it did not impact physicians' behavior toward antibiotic prescriptions. Replacing rapid antigen tests with the FilmArray RP introduced on December 1, 2020, did not affect the magnitude of the reduction in antibiotic prescription rate for pediatric respiratory infections.

## Introduction

The COVID-19 pandemic has caused changes in healthcare services. Some studies have reported that the initial phase of the COVID-19 pandemic in 2020 was associated with changes in outpatient antibiotic prescriptions. A Canadian study revealed that the volume of outpatient antibiotic prescriptions and the number of visits due to respiratory infections were

this published article and its Supplementary Information files."

**Funding:** The authors received no specific funding for this work.

**Competing interests:** The authors have declared that no competing interests exist.

significantly reduced with a minimal change in antibiotic prescription rate from the beginning of the pandemic until the end of 2020. This suggests that the reduction in antibiotic prescriptions was potentially due to the reduction in respiratory infections [1]. They also reported that the decrease in outpatient antibiotic volume since the COVID-19 pandemic was higher in children than in adults. Studies investigating the impact of COVID-19 on outpatient antibiotic prescriptions in children are limited. A single-center study in the UK revealed no evidence that antibiotic prescriptions were significantly affected by the COVID-19 pandemic [2]. A US study investigating pediatric outpatient antibiotic prescriptions showed a significant reduction in antibiotic prescription rates since the beginning of the pandemic, followed by an increasing trend until June 2021 [3]. However, evidence regarding whether the reduction in outpatient antibiotics is sustained with the extended duration of the COVID-19 pandemic is limited. Most studies have included data for a year or less since the initiation of the COVID-19 pandemic. Therefore, evaluating the impact of the prolonged COVID-19 pandemic on pediatric outpatient antibiotic prescription is essential to direct future outpatient antibiotic stewardship among children. In addition, the differential impact of the COVID-19 pandemic on pediatric antibiotic prescriptions by country or region is not fully understood.

The FilmArray Respiratory Panel (RP) 2.1 (FilmArray®; Biofire, Salt Lake, UT, USA) is a multiplex polymerase chain reaction (PCR) test used to detect respiratory pathogens (17 viruses and 4 bacteria) [4]. Following approval of the FilmArray RP by the Japanese government, the test was introduced as a usual healthcare practice to detect respiratory pathogens in our center in December 2020. Before its introduction, rapid antigen tests (RATs) were performed for respiratory infections in our center. The impact of FilmArray testing for outpatient antibiotics varies, with limited evidence in children [5–8]. A pediatric study from the US showed an improvement in the appropriate antibiotic treatment rate by comparing one clinic with 298 patients and another with 132 patients [5]. However, further studies with larger sample sizes are required to evaluate the generalizability of these findings.

The primary objective of this study was to evaluate the impact of the prolonged COVID-19 pandemic on outpatient antibiotic prescriptions for pediatric respiratory infections at an acute care hospital in Japan. The study's secondary objective was to investigate the effect of FilmArray RP on pediatric respiratory outpatient antibiotic prescriptions for pediatric respiratory infections.

## Methods

### Study design and setting

This retrospective cohort study evaluated outpatient antibiotic prescriptions for children <20 years in a Japanese acute care hospital from January 2019 to December 2021. We used an interrupted time series analysis (ITSA) model to assess the impact of the COVID-19 pandemic and FilmArray RP on outpatient antibiotic prescriptions. Nara, a prefecture in Japan, declared a state of emergency in April 2020. We established April 2020 through December 2021 as the COVID-19 pandemic period and January 2019 through March 2020 as the COVID-19 pre-pandemic period. FilmArray RP replaced the RAT as a microbiological diagnostic test for pediatric respiratory infections after December 2020. The RAT period for this study was from January 1, 2019, to November 31, 2020, and the multiplex PCR period was from December 1, 2020, to December 31, 2021.

### Data source and ethics

The number of monthly prescriptions for antibiotics was obtained from the electronic medical records of the Nara Prefecture General Medical Center database. Specific informed consent

was waived by the Medical Ethics Committee of Nara Prefectural General Medical Center due to the retrospective nature of the study. In addition, the Medical Ethics Committee of Nara Prefectural General Medical Center has approved the experimental protocol (No. 671). All experiments were performed in accordance with relevant guidelines and regulations.

## Patients and sample

The study included 2,831 patients with cold-like or respiratory symptoms who visited the outpatient clinic at Nara General Medical Center from January 1, 2019, to December 31, 2021. From January 2019 to November 2020, nasopharyngeal samples were collected from patients using nasal BR Swab EN (Tauns Laboratories, Tokyo, Japan) and pharyngeal swabs (Mizuho Medy, Tokyo, Japan) for the RAT. From December 2020, nasopharyngeal samples were collected from patients using the Copan® FLOQSwabs® UTM® nasopharyngeal sample collection kit (Copan, Brescia, Italy) for the FilmArray RP test. All the samples were tested fresh.

## Rapid antigen test

The test is a rapid immunoassay, which qualitatively detects influenza A and B viruses with ImmunoAce Flu (Tauns), adenovirus with ImmunoAce Adeno (Tauns), RS virus with ImmunoAce RSV NEO (Tauns), human metapneumovirus with ImmunoAce hMPV (Tauns), and *Mycoplasma pneumoniae* with Auto Myco (Mizuho Medy).

## FilmArray system

To analyze the detection rate of viruses and bacteria associated with respiratory diseases in patients, the FilmArray Respiratory Panel 2.1 was used, and multiplex PCR was performed according to the manufacturer's instructions [4]. All specimens were processed individually in a biosafety cabinet, and appropriate personal protective devices such as masks, gloves, and gowns were worn during specimen processing. The FilmArray RP test automatically performs nucleic acid extraction, reverse transcription, nucleic acid amplification, and result analysis in a single assessment (one sample) in approximately 45 min. The FilmArray system automatically analyzed and displayed the results. If each target in the result is indicated by "detected" or "not detected," the test result is valid. If all panel targets are displayed with the result "invalid," one of the internal controls is inadequate.

## Statistical analysis

Patient characteristics were compared using the Mann–Whitney U test, Fisher exact test, or chi-square test. The study conducted the ITSA using a two-intervention model [9]. The first intervention was at the beginning of the COVID-19 pandemic in April 2020, when Nara announced a state of emergency for the first time. The second intervention was the introduction of FilmArray RP in December 2020. The primary outcome was the monthly antibiotic prescription rate (number of patients with any systemic antibiotic prescriptions/100 outpatient respiratory infections with RATs or FilmArray RP test). The level and slope change model were applied to the ITSA to evaluate the immediate and gradual impacts of the two interventions. The level change referred to the RR of the outcome in the post-intervention period compared with the pre-intervention period, whereas the slope change represented the RR of the outcome in a month compared with the previous month. A Poisson regression model was applied to the data without significant overdispersion, whereas the negative binomial model was used in the case of considerable overdispersion. The log of outpatient visits with respiratory infections using RATs or the FilmArray RP test was included as an offset term in the

ITSA with the outcome of the antibiotic prescription rate. All outcomes and variables are presented per month in the ITSA. Given the seasonal impact on respiratory infections and antibiotic prescriptions, the calendar month was included as a variable in the ITSA. In addition, the potential to raise awareness of antibiotic stewardship may have affected physicians' behaviors toward antibiotic prescriptions over time, of which the ITSA evaluated the impact.

ITSA with outcomes of the monthly number of antibiotic prescriptions and the number of patients with RAT or FilmArray RP test was also conducted to evaluate the potential differential impacts on antibiotic prescription and patient visits with respiratory infections. All statistical analyses were performed using Stata 14.1 (StataCorp, College Station, TX, USA).

## Results

During the study period, 50,662 pediatric patients visited our center (21,575 in the pre-pandemic period and 29,087 in the pandemic period). RATs and FilmArray RP tests were conducted for 1,791 and 1,040 cases, respectively. The age range breakdown was 0–3 months, 4–11 months, 1–4 years, 5–9 years, 10–14 years, and 15–20 years, with 6.1%, 13.1%, 52.8%, 19.4%, 7.6%, and 0.9%, respectively, during the pre-pandemic period and 4.9%, 12.1%, 59.0%, 13.7%, 8.7%, and 1.6%, respectively, during the pandemic period. The overall antimicrobial prescription rate decreased from 38.7% during the pre-pandemic period to 22.4% during the pandemic period. Penicillin prescription changed from 3.1% to 1.1%, cephalosporins from 34.2% to 20.1%, quinolones from 0.1% to 0.0%, macrolides from 5.0% to 2.5%, sulfamethoxazole-trimethoprim from 0.1% to 0.0%, and other antibacterials from 0.2% to 0.4%. Thus, a decrease in penicillin, cephalosporins, and macrolides prescriptions was recorded. Patient characteristics are presented in Table 1. As a reference, data by age group are presented in S1 Table online.

The ITSA for the outcome of monthly antibiotic prescription rate revealed a significant decreasing trend in antibiotic prescription rate over the study period (relative risk [RR], 0.95

**Table 1. Patient characteristics and antimicrobial prescription rates during the pre-pandemic and pandemic periods.**

| | Jan 2019 to Mar 2020 | Apr 2020 to Dec 2021 | P value |
|---|---|---|---|
| Average no. of patients per month (min-max) | 1,438 (1,270–1,610) | 1,385 (930–1,823) | 0.511 |
| Average no. of laboratory tests per month (min-max) | 99.9 (63–143) | 63.5 (25–141) | 0.001 |
| Sex % male (min-max) | 53.3 (45.3–61.4) | 54.9 (32.0–73.3) | 0.969 |
| Age % (min-max) | | | <0.001 |
| <3 months | 6.1 (2.7–11.2) | 4.9 (0–12.5) | |
| 3–11 months | 13.1 (4.6–20.0) | 12.1 (3.6–18.9) | |
| 1 year–4 years | 52.8 (39.7–66.3) | 59.0 (44.4–76.7) | |
| 5–9 years | 19.4 (9.5–33.6) | 13.7 (2.7–25.0) | |
| 10–14 years | 7.6 (1.0–14.6) | 8.7 (1.9–19.3) | |
| 15–20 years | 0.9 (0–3.4) | 1.6 (0–4.5) | |
| Antibiotics prescribed % (min-max) | | | |
| Overall antimicrobials | 38.7 (15.9–53.6) | 22.4 (8.3–39.3) | <0.001 |
| Penicillins | 3.1 (0.0–6.7) | 1.1 (0.0–3.8) | <0.0001 |
| Cephalosporins | 34.2 (14.3–47.8) | 20.1 (5.6–35.7) | <0.0001 |
| Quinolones | 0.1 (0.0–1.9) | 0.0 (0.0–0.0) | 0.237 |
| Macrolides | 5.0 (1.1–9.0) | 2.5 (0.0–5.4) | <0.0001 |
| Sulfamethoxazole-trimethoprim | 0.1 (0.0–0.8) | 0.0 (0.0–0.0) | 0.237 |
| Other antibacterials | 0.2 (0.0–1.2) | 0.4 (0.0–2.3) | 0.564 |

**Table 2. Interrupted time series analysis with monthly antibiotic prescription rate, number of antibiotic prescriptions, and number of patient visits with RAT or FilmArray test as outcome.**

| Outcome | RR | Lower 95% CI | Upper 95% CI | P value |
|---|---|---|---|---|
| **Monthly antibiotic prescription rate** | | | | |
| Time | 0.95 | 0.93 | 0.98 | <0.001 |
| COVID-19 | | | | |
| Level | 0.97 | 0.58 | 1.61 | 0.898 |
| Slope | 1.05 | 0.95 | 1.17 | 0.313 |
| FilmArray | | | | |
| Level | 0.90 | 0.46 | 1.75 | 0.753 |
| Slope | 0.95 | 0.85 | 1.06 | 0.330 |
| **Monthly number of antibiotic prescriptions** | | | | |
| Time | 0.97 | 0.94 | 0.99 | 0.003 |
| COVID-19 | | | | |
| Level | 0.35 | 0.21 | 0.57 | <0.001 |
| Slope | 1.01 | 0.91 | 1.12 | 0.888 |
| FilmArray | | | | |
| Level | 1.65 | 0.87 | 3.12 | 0.122 |
| Slope | 1.04 | 0.93 | 1.15 | 0.487 |
| **Monthly number of patient visits with RAT or FilmArray test** | | | | |
| Time | 1.02 | 0.99 | 1.05 | 0.307 |
| COVID-19 | | | | |
| Level | 0.29 | 0.17 | 0.48 | <0.001 |
| Slope | 0.99 | 0.90 | 1.09 | 0.833 |
| FilmArray | | | | |
| Level | 1.51 | 0.79 | 2.86 | 0.208 |
| Slope | 1.06 | 0.96 | 1.18 | 0.223 |

RAT, rapid antigen test; RR, relative risk; CI, confidence interval.

(0.93–0.98) per month; P < 0.001). However, the pandemic (RR level, 0.97 [0.58–1.61]; P = 0.90; RR slope, 1.05 [0.95–1.17] per month; P = 0.310) and FilmArray RP (RR level, 0.90 [0.46–1.75]; P = 0.75; RR slope, 0.95 [0.85–1.06] per month; P = 0.330) (**Table 2**) had no significant impact.

The ITSA for the outcome of the monthly number of antibiotic prescriptions revealed that antibiotic prescriptions significantly decreased over time throughout the study period (RR, 0.97 [0.94–0.99] per month; P = 0.003). Although the level change of the pandemic showed a significant reduction in the monthly number of antibiotic prescriptions, the slope change of the pandemic was not statistically significant (RR level, 0.35 [0.21–0.57]; P < 0.001; RR slope, 1.01 [0.91–1.12] per month; P = 0.890). The FilmArray RP was not related to significant changes in the monthly number of antibiotic prescriptions (RR level, 1.65 [0.87–3.12]; P = 0.12; RR slope, 1.04 [0.93–1.15] per month; P = 0.490) (**Table 2**).

However, the monthly number of patient visits with the RAT or FilmArray test did not show any significant change in trend over time (RR 1.02 [0.99–1.05] per month; P = 0.310). The COVID-19 pandemic had an immediate reduction in the number of patient visits with the RAT and FilmArray test (RR level, 0.29 [0.17–0.48]; P < 0.001; RR slope, 0.99 [0.90–1.09] per month; P = 0.830) (**Table 2**). **Fig 1** shows the trends in the number of patients tested and antibiotic prescription rates during the study period.

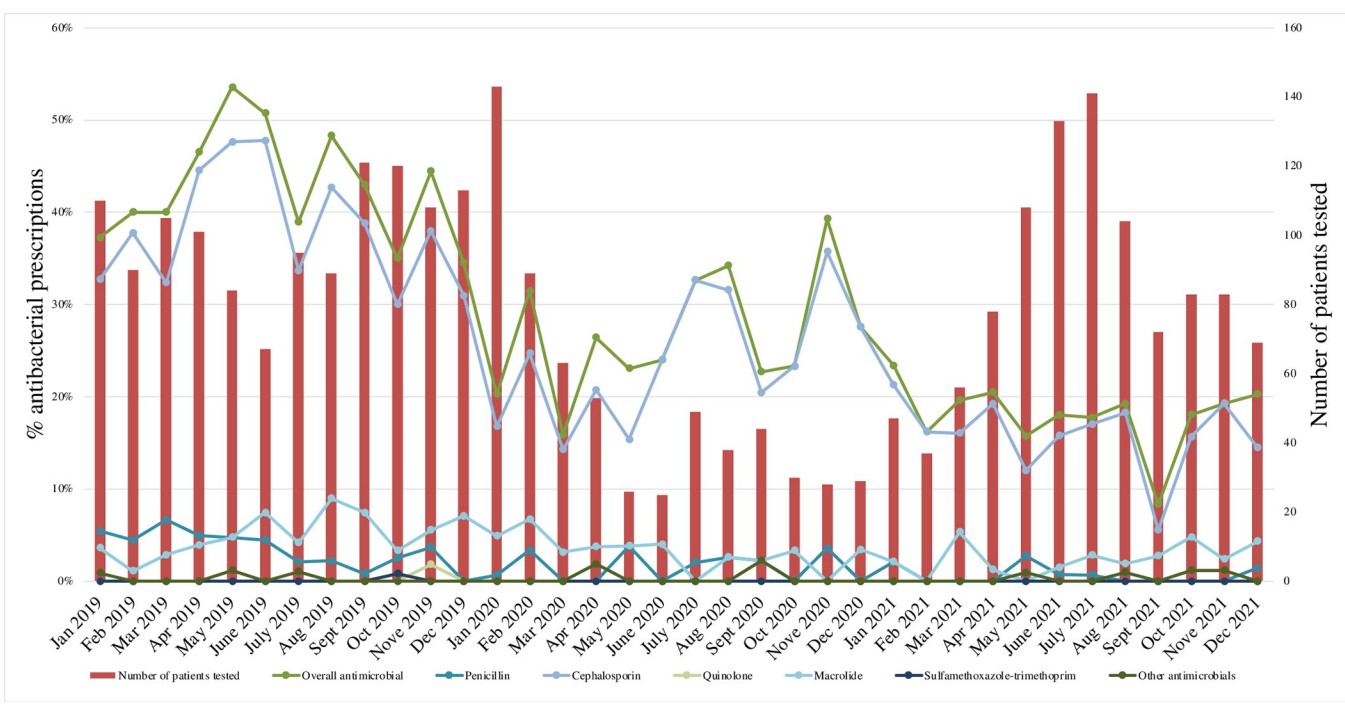

**Fig 1. Trends in the number of patients tested and antibiotic prescriptions, including overall antimicrobials, penicillin, cephalosporins, quinolones, macrolides, sulfamethoxazole-trimethoprim, and other antimicrobials.** Dots indicate the antibacterial prescription rates of these antimicrobials; dots and bars represent the individual monthly totals from January 2019 to December 2021.

## Discussion

This study highlighted the decreasing trend in pediatric antibiotic prescriptions from the beginning of the pandemic through December 2021. Although the COVID-19 pandemic was significantly related to a reduction in the number of antibiotic prescriptions and patient visits with diagnostic tests, this was not associated with the antibiotic prescription rate, suggesting that the pandemic had no impact on physicians' behavior toward antibiotic prescriptions. The FilmArray RP was introduced on December 1, 2020, during the pandemic period. However, replacing the RAT with the FilmArray RP did not affect the magnitude of the reduction in antibiotic prescription rate for pediatric respiratory infections.

Although a US study showed an increasing trend in the antibiotic prescription rate for pediatric outpatient visits until June 2021, our study revealed a sustained reduction until February 2022, almost 2 years since the beginning of the pandemic [3]. The Japanese government published the national action plan in 2016 [10]. Although the awareness of antibiotic stewardship among pediatricians in our center may have been related to the sustained reduction of antibiotic prescription rate during the pandemic period to some extent, other potential factors may have affected the results. For example, our center is a regional reference center where relatively severe patients are referred compared with freestanding walk-in clinics. The change in the severity of referred patients or the threshold for referring clinics to transfer patients to our center may have impacted the results of this study. In addition, this study did not evaluate biochemical laboratory results or radiographic findings, which may have affected the clinician's decision regarding antibiotics prescription.

As reported in previous studies, we also observed an immediate impact of the COVID-19 pandemic on the number of antibiotic prescriptions for respiratory infections and the number

of patient visits with respiratory diagnostic tests, with no changes in the antibiotic prescription rate for respiratory infection [1–3]. This may suggest an increase in the volume of antibiotic prescriptions once the number of patient visits with respiratory infections recovers. Still, the volume of antibiotic prescriptions during the pandemic appears to be lower than that in the pre-pandemic period. Continuous monitoring of the impact of the COVID-19 pandemic on outpatient antibiotic prescriptions is required.

A previous study in our center showed that the use of FilmArray RP testing was associated with a reduction in antibiotic prescriptions for hospitalized children with respiratory infections [11]. However, this outpatient study did not find a significant decrease in outpatient antibiotic prescriptions. This may be due to the severity of respiratory infections between outpatients and inpatients (hospitalized children are more likely to require antibiotic treatment than non-hospitalized children). Another potential explanation for the difference in the results is that the antibiotic prescription rate had been low owing to the impact of the COVID-19 pandemic without room for further improvement, despite our center being a referral center. This outpatient study suggests that the beneficial impact of multiplex PCR testing on the reduction in antibiotic prescriptions for outpatient pediatric respiratory infections is potentially low. In addition, the study did not reveal an additional reduction in antibiotic prescription with the introduction of FilmArray RP, suggesting that non-hospitalized children with respiratory infections may not benefit from the management of antibiotic treatment like hospitalized children. However, these results do not negate the usefulness of the multiplex PCR test for outpatient respiratory infections. The local epidemiology of respiratory infections may be more easily and rapidly monitored using multiplex PCR tests in local healthcare facilities [12, 13].

Our study has some limitations. First, the study was conducted at a single center. As each center may have different antibiotic prescription practices, the impact of the prolonged COVID-19 pandemic on outpatient antibiotic prescriptions may differ by center, country, and region. Further research is warranted to investigate the differential impact of the pandemic on outpatient antibiotic prescription. Second, the ITSA model did not evaluate the impact of patients' characteristics. Additionally, this study did not assess the clinical severity of respiratory infections.

In conclusion, the study showed a sustained reduction in outpatient antibiotic prescriptions for respiratory infections for almost 2 years since the beginning of the prolonged COVID-19 pandemic. Continuous monitoring of the impact of the COVID-19 pandemic on outpatient antibiotic prescriptions is essential.

## Supporting information

**S1 Table. Patient characteristics and monthly antimicrobial prescription rates during the pre-pandemic and pandemic periods.**
(DOCX)

## Author Contributions

**Conceptualization:** Daisuke Kitagawa.

**Data curation:** Madoka Furumori, Soma Suzuki, Yui Shintani, Hiroki Nishikawa, Rika Suzuki, Naohiro Yamamoto, Masayuki Onaka, Atsuko Nishiyama, Takehito Kasamatsu, Naoyuki Shiraishi, Yuki Suzuki, Akiyo Nakano, Ryuichi Nakano.

**Investigation:** Daisuke Kitagawa.

**Methodology:** Daisuke Kitagawa, Taito Kitano.

**Project administration:** Hisakazu Yano, Koichi Maeda, Sayaka Yoshida, Fumihiko Nakamura.

**Supervision:** Hisakazu Yano, Koichi Maeda, Sayaka Yoshida, Fumihiko Nakamura.

**Writing – original draft:** Daisuke Kitagawa, Taito Kitano.

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
