## [Decision Letter · Decision Letter 0]

31 Oct 2022

PONE-D-22-28098Impact of the COVID-19 pandemic and multiplex polymerase chain reaction test on outpatient antibiotic prescriptions for pediatric respiratory infectionPLOS ONE

Dear Dr. Kitagawa,

Thank you for submitting your manuscript to PLOS ONE. After careful consideration, we feel that it has merit but does not fully meet PLOS ONE’s publication criteria as it currently stands. Therefore, we invite you to submit a revised version of the manuscript that addresses the points raised during the review process.

We look forward to receiving your revised manuscript.

Kind regards,

Benjamin M. Liu, MD, PhD

Academic Editor

PLOS ONE

Reviewers' comments:

Reviewer's Responses to Questions

**Comments to the Author**

1. Is the manuscript technically sound, and do the data support the conclusions?

Reviewer #1: Yes

2. Has the statistical analysis been performed appropriately and rigorously? 

Reviewer #1: Yes

3. Have the authors made all data underlying the findings in their manuscript fully available?

Reviewer #1: Yes

4. Is the manuscript presented in an intelligible fashion and written in standard English?

Reviewer #1: Yes

5. Review Comments to the Author

Reviewer #1: This manuscript makes clever use of interrupted time series analysis to analyze the impact of the COVID-19 pandemic and the FilmArray Respiratory Panel on antibiotic prescriptions of the outpatient with pediatric respiratory infections.

Although two time points of the events and their response strategy have no significant correlation with the sustained reduction of antibiotic prescription rate, it still suggests that drug resistance monitoring is as important as epidemic monitoring, and standard guidelines for clinicians' prescribing behavior are more important.

I was very glad to read the interesting findings observed from author’s clinical perspective, The following questions are purely out of personal curiosity and do not affect the logic of the full text.

1. Compared with the pre pandemic period, age rate of patients less than 3 months is declining in pandemic period shown in Table 1. However, antibiotics prescribed rate of cephalosporins increases from 8.5 to 9.3 in Supplementary Table S1. Is this increase statistically significant? If so, why does this phenomenon occur in the youngest infants?

2. The RAT period has crossed pre- and pandemic period, however, the multiplex PCR period only covers pandemic period. Whether the data in the post-pandemic period still support the observation results in this paper is very promising.

In addition, there are some specific details that may need to be modified.

3. Only one P value (=0.33) has two decimal places reserved in Table 2, the others are all three. However, from abstract to text, most P values tend to be two digits on Line 37, 38, 180, 181, 193, 195, 197, and 200 etc., and three on Line 190. Although understanding the author may be due to the simple habit, it could keep consistent with the full manuscript including tables to avoid unnecessary conversion, comparison and misunderstanding for readers.

4. Line 261 to 264 was more suitable for discussion than conclusion. After all, there is no data on hospitalized children in this article, and no comparison to support this conclusion.

5. The interrupted time series analysis appears on Line 177, but its abbreviation ITSA first appeared on Line 87.

6. PLOS authors have the option to publish the peer review history of their article (what does this mean?). If published, this will include your full peer review and any attached files.

Reviewer #1: No

---

## [Author Response · Author response to Decision Letter 0]

9 Nov 2022

Dear Editor and Reviewers:

We thank you for reviewing our manuscript and for providing constructive comments and suggestions. Our responses to the comments are provided below.

1. Compared with the pre pandemic period, age rate of patients less than 3 months is declining in pandemic period shown in Table 1. However, antibiotics prescribed rate of cephalosporins increases from 8.5 to 9.3 in Supplementary Table S1. Is this increase statistically significant? If so, why does this phenomenon occur in the youngest infants?

Author response 1: We thank you for your suggestion. The rate of cephalosporin antibiotic prescriptions for patients younger than 3 months was not significantly different between the pre-pandemic and pandemic periods with a p-value of 0.871. In addition, the use of antimicrobials for patients under 3 months differs from other age groups, even for simple fevers, and the risk of severe illness requiring antimicrobial treatment should be considered in this age group. We believe that it is difficult to evaluate the changes in antimicrobial use for this age group in the same way as for other age groups.

2. The RAT period has crossed pre- and pandemic period, however, the multiplex PCR period only covers pandemic period. Whether the data in the post-pandemic period still support the observation results in this paper is very promising.

Author response 2: We thank you for pointing this out. We will also analyze the Multiplex PCR data during the pandemic period as a future issue.

3. Only one P value (=0.33) has two decimal places reserved in Table 2, the others are all three. However, from abstract to text, most P values tend to be two digits on Line 37, 38, 180, 181, 193, 195, 197, and 200 etc., and three on Line 190. Although understanding the author may be due to the simple habit, it could keep consistent with the full manuscript including tables to avoid unnecessary conversion, comparison and misunderstanding for readers.

Author response 3: We thank you for pointing this out. We have modified all decimal places to three.

4. Line 261 to 264 was more suitable for discussion than conclusion. After all, there is no data on hospitalized children in this article, and no comparison to support this conclusion.

Author response 4: We thank you for your suggestion. We have changed this statement to discussion (Lines 248–252).

5. The interrupted time series analysis appears on Line 177, but its abbreviation ITSA first appeared on Line 87.

Author response 5: We thank you for pointing this out. We have modified the omitted text sections (Lines 88 and 178).

---

## [Editor Report · Decision Letter 1]

24 Nov 2022

Impact of the COVID-19 pandemic and multiplex polymerase chain reaction test on outpatient antibiotic prescriptions for pediatric respiratory infection

PONE-D-22-28098R1

Dear Dr. Kitagawa,

We’re pleased to inform you that your manuscript has been judged scientifically suitable for publication and will be formally accepted for publication once it meets all outstanding technical requirements.

Kind regards,

Benjamin M. Liu, MBBS, PhD, D(ABMM), MB(ASCP)

Academic Editor

PLOS ONE
---

## [Editor Report · Acceptance letter]

23 Dec 2022

PONE-D-22-28098R1 

Impact of the COVID-19 pandemic and multiplex polymerase chain reaction test on outpatient antibiotic prescriptions for pediatric respiratory infection 

Dear Dr. Kitagawa:

I'm pleased to inform you that your manuscript has been deemed suitable for publication in PLOS ONE. Congratulations! Your manuscript is now with our production department. 

Kind regards, 

on behalf of

Dr. Benjamin M. Liu 

Academic Editor

PLOS ONE